# RJ-RRT: Improved RRT for Path Planning in Narrow Passages

Qisen Chai [ID], Yujun Wang *

School of Computer and Information Science, Southwest University, Chongqing 400715, China
* Correspondence: wangyjun@swu.edu.cn

**Abstract:** As a representative of sampling-based planning algorithms, rapidly exploring random tree (RRT), is extensively welcomed in solving robot path planning problems due to its wide application range and easy addition of nonholonomic constraints. However, it is still challenging for RRT to plan the path for configuration space with narrow passages. As a variant algorithm of RRT, rapid random discovery vine (RRV) gives a better solution, but when configuration space contains more obstacles instead narrow passages, RRV performs slightly worse than RRT. In order to solve these problems, this paper re-examines the role of sampling points in RRT. Firstly, according to the state of the random tree expanding towards the current sampling point, a greedy sampling space reduction strategy is proposed, which decreases the redundant expansion of the random tree in space by dynamically changing the sampling space. Secondly, a new narrow passage judgment method is proposed according to the environment around of sampling point. After the narrow passage is identified, the narrow passage is explored by generating multiple subtrees inside the passage. The subtrees can be merged into the main tree that expands in a larger area by subsequent sampling. These improvements further enhance the value of sampling points. Compared with the existing RRT algorithms, the adaptability for different environments is improved, and the planning time and memory usage are saved.

**Keywords:** path planning; RRT; narrow passage; complex environment

## 1. Introduction

With the rising use of applications such as unmanned aerial vehicles (UAV), autonomous driving, and mobile robots, path planning as an essential part of their technology have long been a research hotspot of scholars [1,2]. The main task of path planning is to find an optimal or sub-optimal path from the initial configuration to the goal configuration in the configuration space according to specific performance criteria, such as path length, planning time, and whether the vehicle kinematics are satisfied.

Among many types of algorithms, the graph-based search method represented by A* [3] and Dijkstra's algorithm [4] divides the configuration space mapped from the real environment into multiple grids according to the size of the resolution and then uses the graph search method to find feasible paths. The resolution size has a decisive impact on this type of algorithm's performance. An overly complex and sizeable real environment often causes an exponential increase in computing time and memory usage [5]. As another important class of planning algorithms, sampling-based planning algorithms, represented by RRT [6] and Probabilistic Road Map (PRM) [7], simplify the configuration space by random sampling. Therefore, they do not require complex geometric modelling and perform better in high dimensions or large environments. Probabilistic completeness guarantees that if at least one feasible path exists, the probability of this class of methods finding the path will be 1 [8,9].

Although sampling-based methods have many advantages for path planning, the performance of such algorithms still has challenges when the configuration space contains multiple narrow passages [10,11]. As a single-query, RRT plans a path by constructing a random tree extending from the initial configuration to the goal configuration by continuous

random sampling. As a multi-query, the PRM needs to generate several random sampling points at the preprocess stage and then connect these points to each other to form a connectivity graph, thus capturing the configuration space's connectivity. Due to the special property of random sampling, both RRT and PRM are affected by the volume limitation of free space in the random sampling process. Specifically, the branches generated when RRT constructs a random tree are almost impossible to expand in narrow passages, and PRM is difficult to generate random sampling points in the collision-free configuration space with small volumes, thus affecting the construction of connected graphs. To solve the problem that the sample-based algorithm has insufficient planning ability for narrow passages, scholars have given many solutions. For PRM, in [12], the author proposes an extended-retract map that makes the sampling points all fall on the central axis of free space. In [13], the author proposed a strategy based on Gaussian sampling. This strategy will increase the sampling points located at the boundary between the obstacle and the free space and eventually generate many sampling points around the obstacle. However, it is not considered that the sampling point is around the obstacle but not necessarily inside the narrow passage. In [14], the author proposes a bridge connection test method to detect narrow passages. In this method, a bridge is built by sampling points located on two different obstacles. If the midpoint of the bridge is in free space, it means that it has passed the bridge test. Obviously, narrow passages make it easier to build such bridges. In [15], the author uses an artificial potential field to make the sampling point located on the obstacle move to free space under the action of repulsive force. In [16], the authors coordinately map both the obstacle space and the free space at the same time, increasing the connectivity of the graph inversely through failed connections.

For RRT, since RRT is more goal-oriented than PRM, it is planning a path while constructing a tree, which makes the improvement of RRT have greater potential. In [17], the authors propose a Voronoi bias-based planner and improve the sampling strategy according to the visibility of the regions by nodes in the tree. In [18], the authors regress the sampling points located on the obstacle so that the tree can expand around the edge of the obstacle, thus ensuring the exploration inside the passages. In [19], the author reduces the total number of samples by rejecting nodes located in the repeated sampling area and proposes a random steer strategy for the narrow passage area, as well as the fusion and adjustment of the path to make the path close to the optimal path. In [20], the authors use a simple heuristic to generate additional subtrees in the configuration space and control the expansion of the subtrees through parameters. In [21], a modified bridge test is used to identify narrow passages and relies on multiple subtrees to explore narrow passages. In [22], the author balances the exploration of global and local random trees through Markov chains. In [23], the authors use PRM globally and rely on special sampling points to activate RRT and use RRT to explore the complex environment. Based on this idea, RRT can be combined with various variants of PRM.

In particular, with the development of machine learning methods in recent years, learning-based path-planning methods have been extensively developed. In [24], for dynamically changing environments, the authors build a predictive model using Support Vector Machine (SVM) to capture the connectivity of narrow passages. In [25], the authors learn by analyzing the density of samples around special regions, using semantic information to guide the distribution of sampling points in random sampling. In [26], the authors model the expansion of random trees as a multi-armed bandit problem and learn action values through reinforcement learning. In [27], the author uses a large number of asymptotically optimal paths planned by A* as a dataset to train convolutional neural network (CNN) model, which can predict the optimal path on the map and guide the distribution of sampling points. In [28], the authors perform local sampling around the sampling point that are difficult to expand, and by performing principal component analysis (PCA) on these local sampling points, the local surrounding environment is divided into narrow passage entrances, narrow passage interiors, and obstacle edges, so random trees can climb obstacles in different environments like vines to expand. This method has achieved good

results in the environment with narrow passages. However, due to its dependence on the environment, the performance is slightly worse than RRT in an environment with many obstacles but no narrow passages.

In addition, when solving the problem of narrow passages, the above methods mostly ignore the randomness of whole configuration space sampling, which often leads to part of sampling points in space that are not helpful for the final path construction so that increasing the total planning time and wasting memory resources. This paper proposes a new planning algorithm based on the RRT named Reduce-Judge RRT (RJ-RRT). Specifically, the new algorithm includes two improvements: firstly, based on the greedy idea, a sampling strategy is proposed to reduce the sampling space to the goal area gradually. Secondly, a new method of environment judgment is proposed, when a narrow passage is identified, it can be explored by subtrees expansion inside the passages. The improved environment judgment method only needs a straightforward calculation to determine the environment around the current sampling point. Both strategies improve planning efficiency. On the one hand, random sampling points are used to reduce the sampling space to avoid random tree expansion in space that does not help with path planning. On the other hand, random sampling points are used to detect the surrounding environment, and the subtree of narrow passages helps the random tree expand in the configuration space that is difficult to explore. The results show that the algorithm has good adaptability to various environments, and can greatly reduce the planning time and memory usage.

The rest of this paper is outlined as follows. Section 2 formally defines the path planning problem and introduces the preconditions of the proposed algorithm. Section 3 explains the RJ-RRT method proposed in this paper. Section 4 provides the results and evaluation of the simulation experiments. Section 5 concludes this paper and discusses future research directions.

## 2. Background

This section formally defines the problem that path planning needs to solve and introduces algorithms such as Basic-RRT.

### 2.1. Problem Definition

Let $X \in \mathbb{R}^d$ be the the configuration space of the path planning problem, where $d \in \mathbb{N}$, $d \geq 2$. Let $X_{obs} \in X$ be the obstacle space, which is an impassable space, and denotes the obstacle-free space as $X_{free} = X \setminus X_{obs}$, which is a passable space. $x_{init} \in X_{free}$ and $x_{goal} \in X_{free}$ are the initial configuration and the goal configuration, respectively, and the goal region is defined as a circle with radius r: $X_{goal} = \left\{ x \in X \middle| ||x - x_{goal}|| < r \right\}$. A path is defined by continuous function $\sigma : [0, T] \to X_{free}$, and $\sigma(0) = x_{init}$, $\sigma(T) = x_{goal}$. If the path is feasible, then $\sigma(\tau) \in X_{free}$ for all $\tau \in [0, T]$.

The path planning problem is to find a feasible path. Problem 1 defines the feasibility problem of path planning.

**Problem 1 (Feasible Path Planning)** Given a configuration space of planning problem $X \in \mathbb{R}^d$, a free space $X_{free}$, an initial state $x_{init} \in X_{free}$, and a goal region $X_{goal} \in X_{free}$, find a path $\sigma : [0 : T] \to X_{free}$ if one exists. If no such feasible path exists, return failure.

The cost from the initial configuration to the goal region is not necessarily the same each time planning or on each path. Let $c(\cdot)$ be the cost function. The optimization problem of path planning is to find the path with minimum (non-negative) path cost. Its formal expression is given by Problem 2.

**Problem 2 (Optimal Path Planning)** Given a configuration space of planning problem $X \in \mathbb{R}^d$, a free space $X_{free}$, an initial state $x_{init} \in X_{free}$, a goal region $X_{goal} \in X_{free}$ and cost function $c$. Find a feasible path $\sigma^*$, such that $c(\sigma^*) = min \left\{ c(\sigma) : \sigma \in \Sigma_{feasible} \right\}$.

The path planning algorithm usually takes some time to plan the path. Let $t \in \mathbb{R}$ be the set of times it takes to find a set of paths. Problem 3 defines finding the best path in Problem 2 in the least amount of time possible.

**Problem 3 (Fast Path Planning)** Minimum time $t \in \mathbb{R}$ required to plan the Optimal Path.

*2.2. Basic-RRT*

The Basic-RRT algorithm is described in Algorithm 1 and the Basic-RRT is the basis of RJ-RRT, an underlying tree data structure is maintained in Basic-RRT. As shown in Figure 1. Firstly, Basic-RRT creates a tree rooted at the $x_{init}$ (Line 1 in Algorithm 1). In each subsequent iteration, $x_{rand}$ is obtained by randomly sampling in the configuration space $X$, traversing the existing nodes in the tree, selecting the node $x_{near}$ closest to $x_{rand}$, and generating $x_{new}$ through the steering function with a certain expand size (Lines 3–5 in Algorithm 1). If the edge $\{x_{near}, x_{new}\}$ is not located in $X_{obs}$, then $x_{new}$ is added to the tree with $x_{near}$ as the parent node, and the edge $\{x_{near}, x_{new}\}$ will be recorded (Lines 6–8 in Algorithm 1). When $x_{new}$ is within the range of $X_{goal}$, the planning is ended, and the tree is returned. The return fails when the time-out or the number of iterations is exceeded (Lines 9–11 in Algorithm 1).

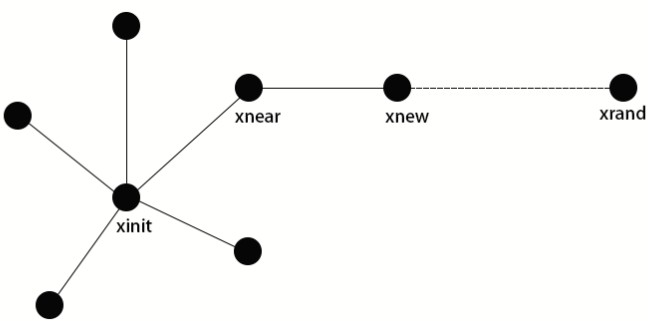

**Figure 1.** Expand of Basic-RRT.

---

**Algorithm 1** Basie-RRT

---

  1:   $Tree \leftarrow x_{init}$
  2: **for** $i = 1$ to $n$ **do**
  3:     $x_{rand} \leftarrow Sample()$
  4:     $x_{near} \leftarrow Nearest()$
  5:     $x_{new} \leftarrow Steer(x_{rand}, x_{near})$
  6:     **if** $FreeCollision(x_{rand}, x_{near})$ **then**
  7:       $Tree.addEdge(x_{near}, x_{new})$
  8:       $Tree.addNode(x_{new})$
  9:       **if** $x_{new} \in X_{goal}$ **then**
10:         return *Tree*
11:       **end if**
12:     **end if**
13: **end for**

---

RRT-Biased [29] is an efficient method to improve RRT. By the premise of a certain probability, the $x_{goal}$ is set as the sampling point so that the tree can expands faster to the goal area. In Algorithm 2, given a brief description of RRT-Biased.

**Algorithm 2** Biased-RRT

---

1: $prand \leftarrow RandomNumber()$
2: **if** $prand \leq K$ **then**
3:     $x_{rand} = x_{goal}$
4: **else**
5:     $x_{rand} \leftarrow Sample()$
6: **end if**

---

In this paper, the tree is rooted at $x_{init}$ denoted as main tree. The rest of the trees denoted as subtrees.

### 3. RJ-RRT

In the Basic-RRT algorithm, the selection of random sampling points is in the entire configuration space or can be described as blind, such as in a certain sampling, the current sampling point is likely very close to the goal region, however, due to the randomness of sampling, the position of the sampling point in the subsequent sampling process is biased., the random tree will likely expand to a region that does not have a facilitation effect on the final path generation. RRT-Biased, although through a forced modification make the random tree is biased expand towards the goal region, when encountering a trap or a complex environment, it falls into the same predicament as Basic-RRT. The role of sampling points for Basic-RRT is to guide the expansion direction of the random tree. If the random tree can quickly approach the goal region, it needs to expand toward the goal region while avoiding obstacles. This is also a requirement for an efficient planner can generate sampling points. To achieve this efficacy and enhance the value of sampling points, in RJ-RRT, sampling space greedy reduction and environment judgement are performed through sampling points, and subtrees are expand in special environments. The overall framework of the RJ-RRT is given by Algorithm 3 and in the subsequent sections detailed description of each improvement.

**Algorithm 3** RJ-RRT

---

1: $MainTree \leftarrow x_{init}$
2: **for** $i = 1$ to $n$ **do**
3:     $x_{rand} \leftarrow GreedSpaceSampling()$
4:     $SubTrees \leftarrow EnvironmentalJudgment(x_{rand})$
5:     **if** $FreeCollision(x_{rand}, x_{near})$ **then**
6:         $MainTree.addEdge(x_{near}, x_{new})$
7:         $MainTree.addNode(x_{new})$
8:         **if** $x_{new} \in X_{goal}$ **then**
9:             return $MainTree$
10:        **end if**
11:        $SubTreeExpand()$
12:        $SubTreeMerge()$
13:    **end if**
14: **end for**

---

#### 3.1. Sampling Space Greedy Reduction Strategy

When there is only the root node $x_{init}$ in the tree, like the sampling strategy is the same as Basic-RRT, sampling points will be randomly selected in the entire configuration space $X$ (Line 1 in Algorithm 3). A space greedy reduction strategy is executed when $x_{new}$ is successfully added to the tree, which is shown in Algorithm 4. For ease of description, the configuration space is denoted as the first $OriginalSpace$. When a node $x_{new}$ is successfully added to the tree, the $OriginalSpace$ will be reduced (Lines 1–2 in Algorithm 4). The specific process is to take $x_{new}$ as the boundary point of the new sampling space and use the boundary point as the benchmark to divide the $OriginalSpace$ in each dimension to obtain

a subspace containing the goal region. This subspace will be used as the space for the next random sampling and denoted as *NextSpace*. In the next sampling, this *NextSpace* will replaced by *OriginalSpace* (Lines 3–6 in Algorithm 4). At this time, the sampling space completes a reduction. After each sampling space is reduced, record the *GapSpace* between *NextSpace* and the *OriginalSpace*, where *GapSpace* is defined as Equation (1):

$$GapSpace = OriginalSpace \setminus (OriginalSpace \cap NextSpace) \tag{1}$$

As the number of space reductions increases, will generate $N$ *GapSpace*, denoted as $Gap_1, Gap_2 \ldots Gap_N$. Ideally, the sampling space will gradually approach the goal region as the number of sampling increases, but this situation is often not practical. When the random tree expansion with random sampling fails in the *OriginalSpace*, it means that the current space has failed to be reduced, and the space to sample will fall back to the *GapSpace* ($Gap_N$) nearest to the *OriginalSpace* (Line 8 in Algorithm 4). Specifically, when executing this function, sampling will first take place in the *GapSpace* and perform multiple times to provide more node expansion options for the next space reduction. When the specified sampling times are reached in the *GapSpace*, forward to the space where the space reduction failed, that is, the *OriginalSpace* for sampling. If the expansion is successful after sampling, the random tree continues to expand according to the above reduction strategy. If the expansion fails, fall back to the previous *GapSpace* ($Gap_{N-1}$) sampling of the *GapSpace* ($Gap_N$). After sampling multiple times, forward to the *GapSpace* ($Gap_N$), sample again, and finally, forward to the *OriginalSpace* for sampling. If the reduction still fails, continue to fall back to sampling in more outer *GapSpace*, and then sample in the next *GapSpace* of each *GapSpace* in turn. When fall back to the first *GapSpace* sampling, namely $Gap_1$, the random tree generated by RJ-RRT in the configuration space is like the Basic- RRT.

---

**Algorithm 4** GreedSpaceSampling

---

1: $OriginalSpace = ConfigurationSpace$
2: $x_{rand} \leftarrow Sample(OriginalSpace)$
3: **if** $FreeCollision(x_{rand}, x_{near})$ **then**
4:     $NextSpace \leftarrow Reduce(x_{rand}, X_{goal})$
5:     $GapSpace \leftarrow Reduce(OriginalSpace, NextSpace)$
6:     $OriginalSpace = NextSpace$
7: **else**
8:     $x_{rand} \leftarrow BackwithForwardSample(GapSpace)$
9: **end if**

---

Take Figure 2 as an example. Firstly, $A$ is the configuration space and the first *OriginalSpace*. After the sampling and steering function, the newly added node is $P1$. The sampling space is reduced as the random tree expands, through $P1$ formed new sampling space $B$. Sampling in $B$ to obtain node $P2$, the space is further reduced to obtain sampling space $C$, sampling in $C$, random tree expansion is difficult, indicating that the current space cannot be further reduced, fall back to the *GapSpace* between $B$ and $C$ for sampling, and after multiple samplings, forward to sample in $C$, the space reduce still fail, come back to sampling in the *GapSpace* between $A$ and $B$ to sample multiple times, forward again to sampling in the *GapSpace* between $B$ and $C$, and finally forward to sampling and expanding in $C$.

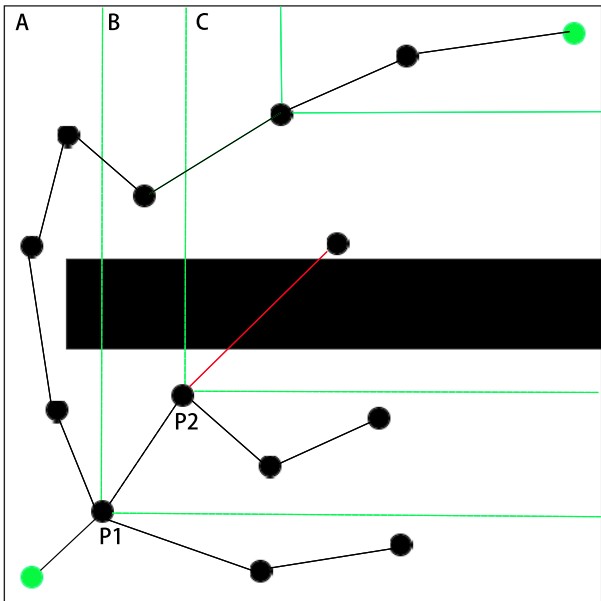

**Figure 2.** The sampling space A is reduced to B through P1 and then reduced to C through P2. The expansion fails in C. The first fall back sampling and then forward sampling.

### 3.2. Environmental Judgment

The main purpose of environmental judgment is to identify narrow passages and set the expansion space of subtrees (Line 4 in Algorithm 3). Environmental judgment will mainly occur during part of the sampling process, and special cases where environmental judgment is not performed will be described in the next section.

When randomly generating a sampling point $x_{rand}$ in the sampling space, the environment around $x_{rand}$ will be judged. As shown in Figure 3. Firstly, if $x_{rand}$ is located in $X_{free}$, the subsequent judgment will be omitted. If $x_{rand}$ is located in $X_{obs}$, then perform local sampling in the circle with $x_{rand}$ as the center and $R1$ as the radius to obtain $N1$ sampling points and determine the space where these sampling points are located (Line 1 in Algorithm 5). If there are sample points located in $X_{free}$, it can be inferred that $x_{rand}$ is at the boundary of the obstacle and close to free space. Randomly pick a sampling point in $X_{free}$, denoted as $x_{cir}$. Divide the circle with $x_{cir}$ as the center and $R2$ as the radius into eight equal parts and obtain eight marked points on the boundary of the circle, denoted as $M1$, $M2$, $M3$, $M4$, $M5$, $M6$, $M7$, and $M8$ (Lines 2–3 in Algorithm 5). Through the location of the marked points, it is further identified whether the current surrounding environment of $x_{cir}$ is the inside of the narrow passage or the entrance of the narrow passage. The process of judging the surrounding environment of the sampling point is given by Algorithm 5.

---

**Algorithm 5** EnvironmentalJudgment

---

1: $x_{cir} \leftarrow LocalSample(x_{rand})$
2: $MarkedPoints \leftarrow LocalJudgement(x_{cir})$
3: $SubTreeExpandSpace \leftarrow LocalJudgement(MarkedPoints)$

---

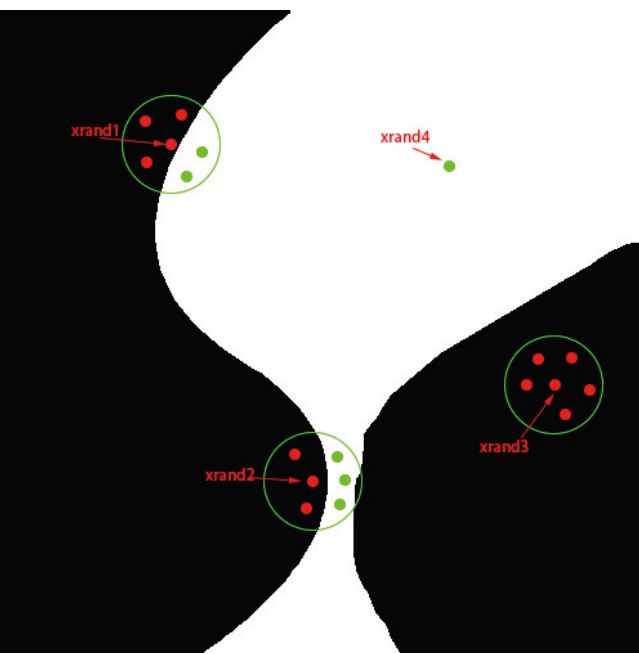

**Figure 3.** Locally sample around *xrand*. *xrand*1, *xrand*2, *xrand*3, and *xrand*4 show four different cases, respectively. When there are sampling points in the free space in the local sampling, such as *xrand*1 and *xrand*2, then the surrounding environment of *xrand* is further judgments, *xrand*3 and *xrand*4 will not make further judgments.

**Inside the narrow passage**: as shown in Figure 4, when two or four marked points are located in $X_{free}$, the remaining marked points are located in $X_{obs}$, and where each marked point located in free space is matched with another free point with an angle of 180°. Then, we inferred that $x_{cir}$ is located inside the narrow passage. In this case, the constructed subtree expansion space is a rectangle (If the configuration space is 3D, will be a cube). The center of the rectangle is at $x_{cir}$, where the long side is parallel to the line formed by the 180° marked points in free space, and the length of the side is $L1$. The other short side of the rectangle is $L2$. This stretched rectangle constitutes the sampling space of the subtree.

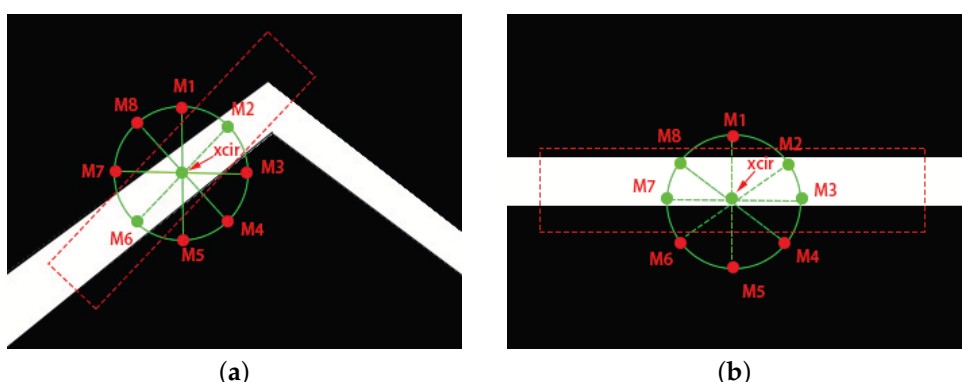

| (a) | (b) |
|-----|-----|

**Figure 4.** (**a**,**b**) are examples of $x_{cir}$ inside the narrow passage.

**Entrance of the narrow passage**: when the marked points are distributed similarly to Figure 5, that is, there are three or four consecutive marked points and an additional isolated marked point (in Figure 5, the isolated marked point is M3) in the free space, while the adjacent to the isolated marked point or remaining marked points are located in the obstacle space then it is inferred that $x_{cir}$ is located at the entrance of the narrow passage, and isolated marked point is used to identify narrow passage entrance and guide exploration. The constructed subtree expansion space is like that inside the narrow space.

The difference is that the stretching of the subtree expansion space is only carried out in the same direction as the isolated marked point.

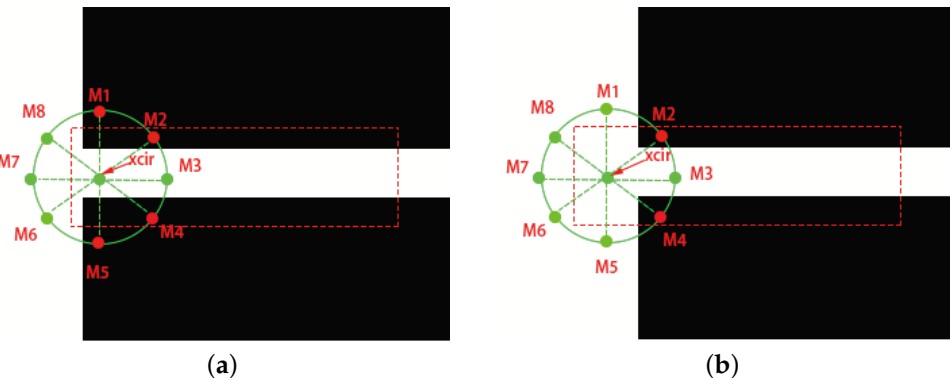

**Figure 5.** (**a**,**b**) are examples of $x_{cir}$ in the entrance of the narrow passage.

The generated subtree expansion space roughly depicts the outline of the narrow passage without the details of the narrow passage configuration, so this method does not require complex calculations. Using randomness of RRT sampling can obtain the configuration of free space in subtree extended space. Once a subtree expansion space is determined according to $x_{cir}$, the subtree expansion will be performed immediately. There are two attendant problems. Question 1: How to avoid the unlimited generation of multiple $x_{cir}$? Question 2: How to avoid repeated exploration of the same narrow passage by multiple subtrees due to overlapping subtree expansion spaces? For Question 1, the unrestricted generation of subtrees is prevented by judging whether the $x_{cir}$ is already within the expansion interval of some subtree. This method will also cause problems raised in Question 2. For Question 2, it can be solved by simple subtree merging, and the specific measures are discussed in the next section.

### 3.3. Subtree Expansion and Merge

#### 3.3.1. Subtree Expansion

When judging that $x_{cir}$ is located inside a narrow passage or at the entrance, take $x_{cir}$ as the root of the subtree, randomly sample, and expand in the constructed subtree expansion space (Line 11 in Algorithm 3), and the expansion method is the same as that of Basic-RRT. Figure 6 shows the flow of subtree expansion. In addition, the expansion of the subtree will have a pre-expansion process, firstly sampling N2 times in the subtree expansion space. These sampling points do not make environmental judgments, and the external main tree does not expand with these sampling points. For other subtrees, if the distance between the $x_{near}$ and $x_{rand}$ is less than $D1$, it indicates that these subtrees may be very close, and these subtrees will expand with this sampling. Then, when this space sampling ends, it will return to the *OriginalSpace* for sampling, and both the main tree and the subtrees will expand with each sampling. However the space reduction only happens in the expansion of the main tree.

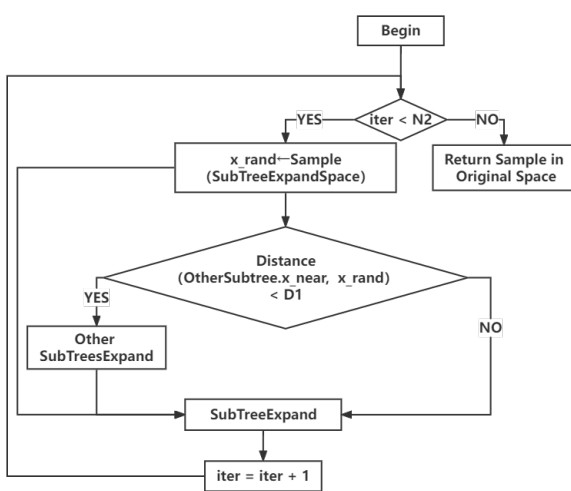

**Figure 6.** Schematic of SubTree expansion.

### 3.3.2. Subtree Merge

Subtree merge can be divided into subtree with subtree and subtree with main tree merge (Line 12 in Algorithm 3). Subtree process of merging is given by Algorithm 6.

When sampling in $OriginalSpace$, if the distance between the nearest node is less than $D2$ after the main tree and the subtrees expand once, the subtrees are merged into the main tree (Lines 4–5 in Algorithm 6). When sampling in the $OriginalSpace$ or the subtree expansion space, if there are two or more subtrees after expanding once, the distance between the nearest nodes is less than $D2$, then the multiple subtrees are merged into one subtree (Lines 7–9 in Algorithm 6). This method not only merges multiple subtrees but also avoids the over-exploration of narrow passages caused by overlapping subtree expansion spaces.

The premise of merging multiple random trees is that there is no collision between the nearest nodes.

---

**Algorithm 6** Subtree merge

---

1: **if** $x_{rand} \leftarrow Sample()$ **then**
2:    $MainTreeExpand()$
3:    $SubTreeExpand()$
4:    **if** $Distance(MainTree, SubTrees) < D2$ **then**
5:       $MainTree = (MainTree \cup SubTree)$
6:    **end if**
7:    **if** $Distance(SubTree, OtherSubTree) < D2$ **then**
8:       $SubTree = (SubTree \cup OtherSubTree)$
9:    **end if**
10: **end if**

---

## 4. Simulation and Result

In this section, to evaluate the performance of the proposed algorithm, RJ-RRT is compared with Basic-RRT and RRV through a series of experiments. Simulation experiments were tested in three groups of different environments. As shown in Figure 7, three representative maps are designed in two-dimensional space: complex environment (containing many obstacles), narrow passage, and bug trap environment. The map size of each environment is 10*10. The initial configuration in the first two groups is $[0.5, 0.5]$, the last group's initial configuration is $[2, 3.5]$, and the goal configuration uniformly is $[9.8, 9.8]$. The black area represents obstacles, the green line represents the expansion state of the random tree, and the red line represents the final planned path. Each set of experiments is performed 50 times to collect experimental data. In order to clearly describe the differences in the per-

formance of different algorithms, the average and maximum and minimum execution times of different algorithms in 50 experiments were calculated, and the number of generated nodes was used to reflect the memory usage. The number of collision detections for RRV and RJ-RRT includes additional nodes required for environmental judgment. All-time units are seconds. All algorithms use Gilbert–Johnson–Keerthi (GJK) [30] for collision detection, and the maximum number of iterations is 50,000. To speed up the planning, the three algorithms uniformly set the sampling probability in the goal space as 0.1. The expand size in the first two sets of experiments was 0.1. In our RJ-RRT, R1 is set to 0.5, N1 is set to 15, R2 is set to 0.7, L1 is set to 3, L2 is set to 1.5, and D1 and D2 are set to 0.7 and 0.5, respectively. All experiments were run on Windows 10 with Intel I7-7700, 3.6 Ghz 8 GB RAM processor, and using MATLAB 2019b as a software platform.

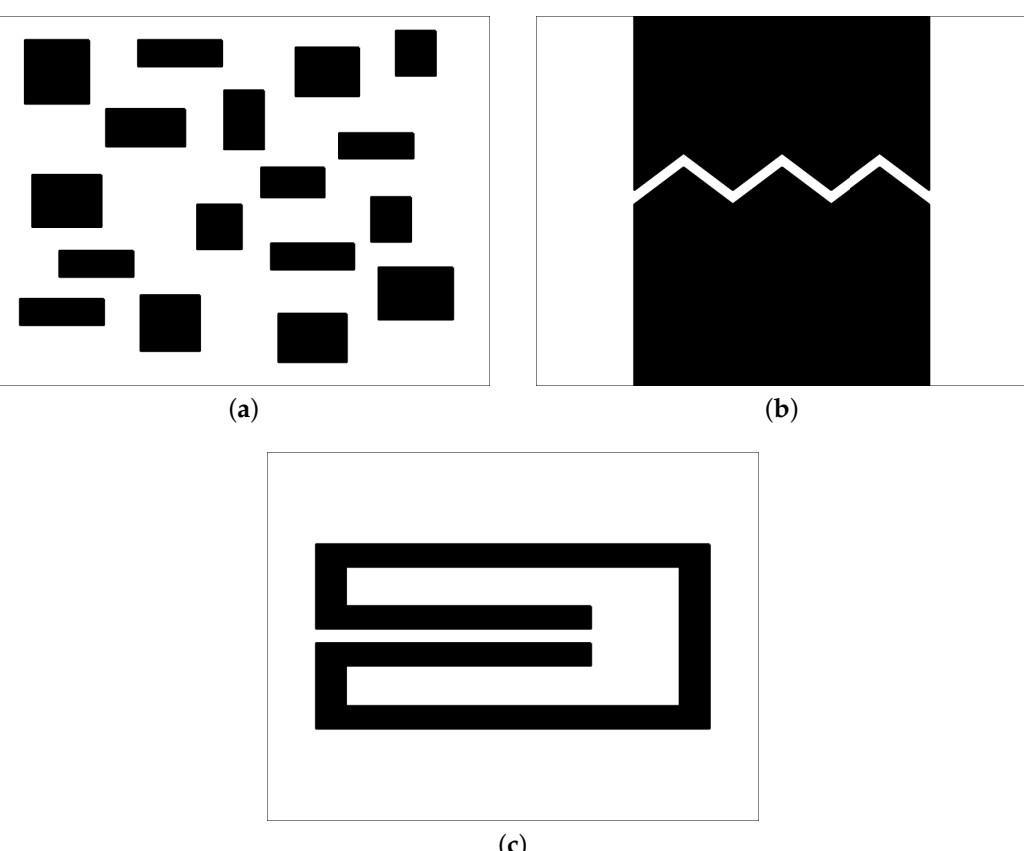

**Figure 7.** Three different maps in the experiment. (**a**) Complex; (**b**) Narrow passage; (**c**) Bug Trap.

*4.1. Complex Environment*

Figure 8 shows the performance of the three algorithms in a complex environment. When testing the Basic-RRT algorithm, due to random sampling in the entire configuration space, the final generated random tree occupies almost the entire space. However, although the final tree generated by the RRV algorithm produces fewer nodes than RRT, it still expands in a lot of useless space, and PCA takes a lot of time to calculate. The average number of samples required by the RRV to plan a path is 29% less than that of the Basic-RRT, but the final average planning time is 51% higher than that of the Basic-RRT. The RJ-RRT has the best performance in this environment. The experimental results in the figure show that only a few redundant nodes are generated during the sampling process, and the planning time is the shortest. The specific statistical results of the three algorithms are shown in Table 1. It can be seen that with the setting of the maximum number of iterations, the three algorithms can find the path 100% in the test. In terms of planning time, RJ-RRT saves

87% compared to Basic-RRT, and 92% compared to RRV. The memory consumption of the generating node is 96% lower than that of Basic-RRT and 94% lower than that of RRV.

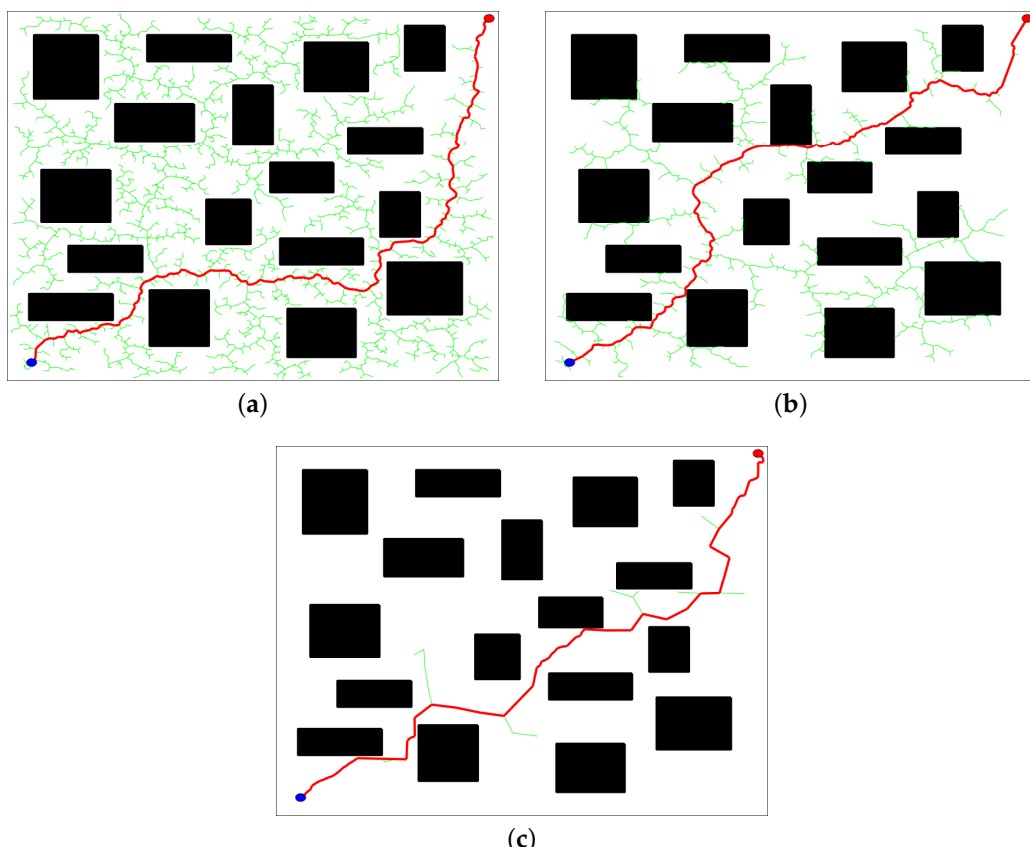

**Figure 8.** The performance of the three algorithms in complex environments. (**a**) RRT; (**b**) RRV; (**c**) RJ-RRT.

**Table 1.** Results of three algorithms in complex environment.

| Algorithm | Average Time (s) | Min Time (s) | Max Time (s) | Average Number of Collisions Detections | Average Number of Nodes |
|---|---|---|---|---|---|
| RRT | 1.982 | 0.452 | 2.741 | 2338 | 1542 |
| RRV | 3.143 | 1.310 | 5.593 | 2975 | 1094 |
| RJ-RRT | 0.239 | 0.127 | 0.874 | 103 | 55 |

*4.2. Narrow Passage*

Figure 9 shows the planning performance of the three algorithms in a narrow passage environment. The experimental results show that when Basic-RRT faces a configuration space containing narrow passages, due to the lack of identification of the particular configuration of the narrow passage, it is not only challenging to find the entrance of the narrow passage but also challenging to expand inside the passage, so successful path-planning usually requires a large number of sampling points and more time. From Table 2, it can be concluded that, in this environment, both RRV and RJ-RRT perform better than RRT, while RJ-RRT requires the least amount of time and takes the least number of samples, on average 1.276 s, and generates an average of 227 sampling nodes. Compared with RRV, it saves 45% of planning time and 76% of the generation of the number of nodes. This means that the RJ-RRT can quickly plan a feasible path and occupy fewer memory resources for a narrow passage environment.

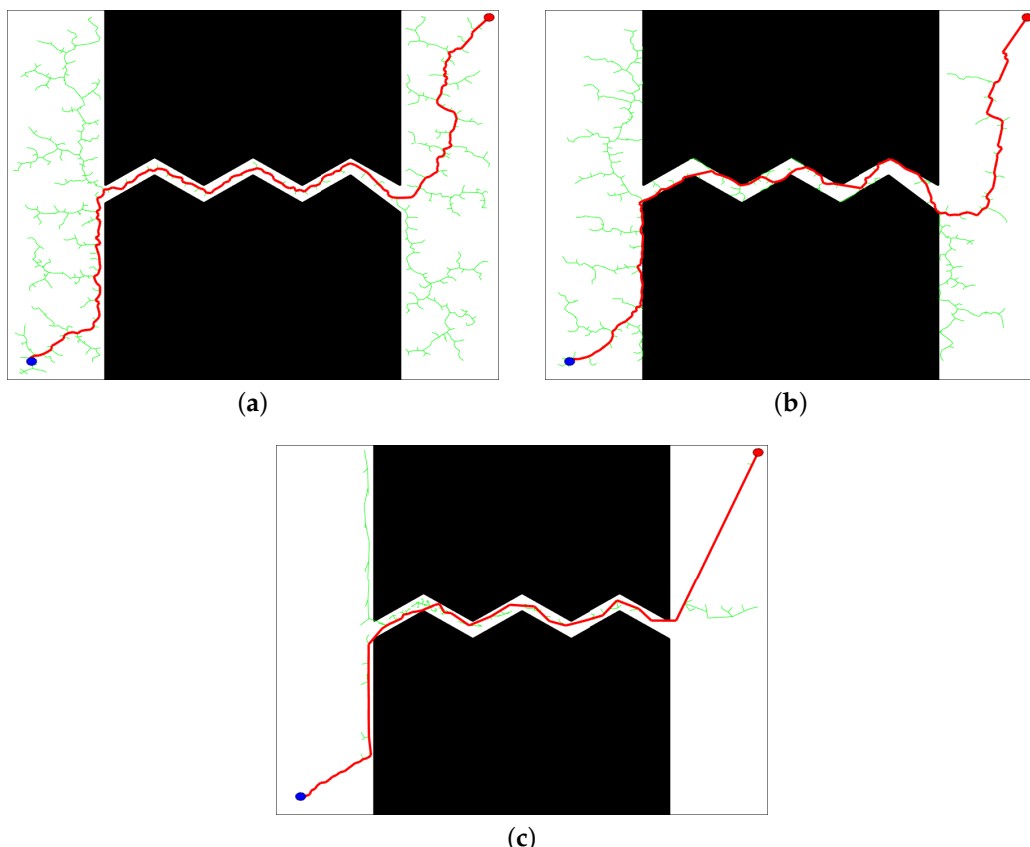

**Figure 9.** The performance of the three algorithms in narrow passage. (**a**) RRT; (**b**) RRV; (**c**) RJ-RRT.

**Table 2.** Results of three algorithms in narrow passage.

| Algorithm | Average Time (s) | Min Time (s) | Max Time (s) | Average Number of Collisions Detections | Average Number of Nodes |
|---|---|---|---|---|---|
| RRT | 4.937 | 1.575 | 10.489 | 7648 | 1347 |
| RRV | 2.338 | 1.031 | 5.944 | 3327 | 958 |
| RJ-RRT | 1.276 | 0.475 | 1.749 | 678 | 227 |

*4.3. Bug Trap*

The expanded size also affects the number of iterations and memory usage of the RRT and RRT variants. In RJ-RRT, the magnitude of each reduction in the sampling space is also determined by the size of the random tree expansion. Therefore, to analyze the influence of different expand sizes on the performance of different algorithms, we use 0.1/0.3 as the expand size for discussion in this experiment. Tables 3 and 4 are the experimental data results when the expand size is 0.1 and 0.3, respectively, and Figure 10 shows the planning performance when the expand size is 0.3. It can be seen from Tables 3 and 4 that RJ-RRT and RRV perform better than RRT in various performance aspects under different expand size settings. When the expand size is 0.1, RJ-RRT is higher than RRV in generating the total number of nodes. This is because the environment in this experimental, the long and narrow trap passage and short expand size lead to multiple *GapSpaces* in the RJ-RRT planning process, resulting in the need for multiple fallback and forward operations. These processes increase the number of samples and planning time, so the final performance is not as good as RRV. When the expand size is set to 0.3, since the expand size increase reduces the number of Gap Spaces, RJ-RRT can still plan a path using a smaller number of nodes and a faster time.

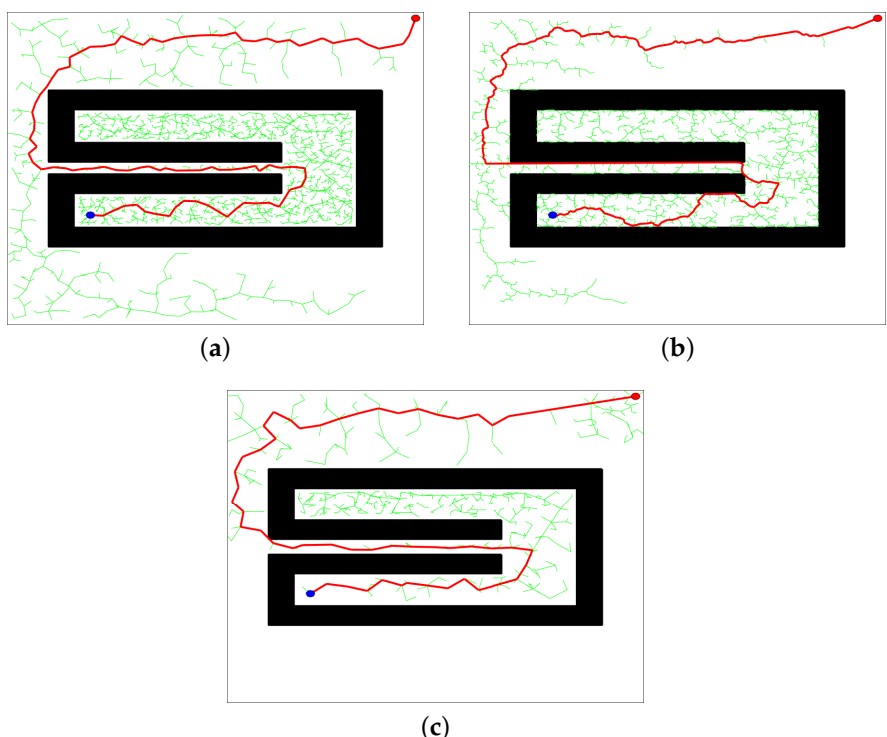

**Figure 10.** The performance of the three algorithms in bug trap. (**a**) RRT; (**b**) RRV; (**c**) RJ-RRT.

**Table 3.** When extended size is 0.1 , the results of three algorithms in the bug trap.

| Algorithm | Average Time (s) | Min Time (s) | Max Time (s) | Average Number of Collisions Detections | Average Number of Nodes |
|---|---|---|---|---|---|
| RRT | 9.846 | 6.959 | 17.707 | 14596 | 5579 |
| RRV | 5.575 | 4.945 | 13.689 | 5920 | 2861 |
| RJ-RRT | 7.194 | 5.746 | 15.145 | 6218 | 3347 |

**Table 4.** When extended size is 0.3, the results of three algorithms in the bug trap.

| Algorithm | Average Time (s) | Min Time (s) | Max Time (s) | Average Number of Collisions Detections | Average Number of Nodes |
|---|---|---|---|---|---|
| RRT | 7.912 | 5.457 | 14.126 | 9829 | 2361 |
| RRV | 4.587 | 3.746 | 10.578 | 5124 | 1741 |
| RJ-RRT | 3.348 | 1.841 | 6.432 | 2546 | 967 |

*4.4. Path Length*

During the experiment, there was an additional discussion, due to the improvement of RJ-RRT's two strategies of greedy space reduction and dependent subtrees exploration, the length of the final planned path is also reduced to a certain extent. Since the main content discussed in this paper is to improve the adaptability of the RRT algorithm in complex environments, especially in the narrow passage, rather than the path length, so only the path length is briefly discussed in this experiment. Figure 7a is used for the map in this experiment, and the rest of the settings are the same as those described above. RRT, RRV and RJ-RRT were tested 50 times, respectively, and the difference between the average of the planned path length of the three algorithms and the shortest path was compared. Figure 11 shows the comparison results of different algorithms on path length. It can be

seen that the difference in length between RRV and RJ-RRT is not obvious, about 82%, 87% different from the shortest path and they both save much more distance than RRT.

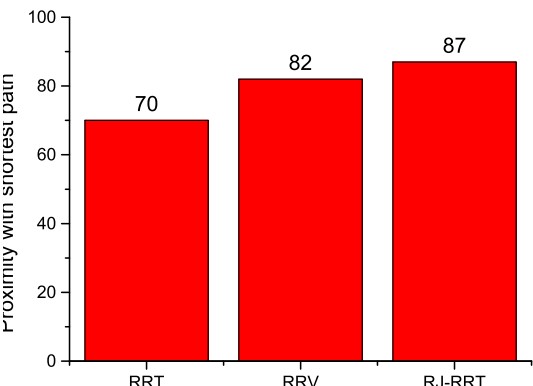

**Figure 11.** Approximation of three algorithms to the shortest path.

## 5. Conclusions

This paper proposes a new planning algorithm RJ-RRT based on the RRT algorithm to solve the planning problem in narrow channels. A new greedy sampling space reduction and environment judgment strategy are proposed in RJ-RRT. While accelerating the expansion of random tree toward the goal area and reducing redundant nodes, it can identify narrow passages and rely on the subtrees expend inside to explore passages.

Compared with RRT and RRV algorithms, RJ-RRT has better adaptability to different environments and maintains a faster planning speed. In addition, another advantage of RJ-RRT is that easy to combine with other RRT variant algorithms. However, the RJ-RRT still has limitations. For example, is easily affected by the expand size. Therefore, in future research, consider combining adaptive expand size and selectively merging multiple Gap Spaces, while RJ-RRT should also be extended to 3D environments and incorporate robot kinematics, making our algorithm closer to practical applications.

**Author Contributions:** Conceptualization, Q.C. and Y.W.; methodology, Q.C.; software, Q.C.; validation, Q.C. and Y.W.; formal analysis, Q.C.; investigation, Q.C.; resources, Q.C.; data curation, Q.C.; writing—original draft preparation, Q.C.; writing—review and editing, Q.C.; visualization, Q.C.; supervision, Y.W.; project administration, Q.C. All authors have read and agreed to the published version of the manuscript.

**Funding:** This research was funded by Chongqing Science and Technology Bureau (cstc2017zdcy-zdyfX0042).

**Institutional Review Board Statement:** Not applicable.

**Informed Consent Statement:** Not applicable.

**Data Availability Statement:** Not applicable.

**Conflicts of Interest:** The authors declare no conflicts of interest.

## Abbreviations

The following abbreviations are used in this manuscript:

| | |
|---|---|
| RRT | Rapidly exploring Random Tree |
| PRM | Probabilistic Roadmap Method |
| RRV | Rapidly exploring Random Vines |
| RJ-RRT | Reduce-Judge RRT |

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
