# Peer review of "RJ-RRT: Improved RRT for Path Planning in Narrow Passages"

_applsci, doi:10.3390/app122312033_

Round 1
Reviewer 1 Report
Well written article, easy to read and enjoyable. Comparison of several methods of path finder and development of new one certainly appreciated.
Modest suggestion is, in some places found use of "we" such as in lines 256, 319, 105, 85 etc..., which is a bit concern; could be in third person passive voice.
Reviewer 2 Report
The proposed algorithm with different method of experiments suggested and experimented is good.
In the Environmental Judgment given the subtrees evaluation and better needs improvement in the different experiments results
Reviewer 3 Report
The paper presents a modified version of the RRT (Rapidly-exploring random tree) method for robot path planning in a static environment. The presented idea might be valuable, but the description of the algorithm need to be enhanced, as well as the explanation of presented results, e.g. why why the solution (red line) exceeds the areas occupied by obstacles in Figure 8 c? Other detailed remarks are given in the attached file.

Reviewer 4 Report
(1) Some of the references cited are of long duration, lack of research significance, and the latest literature is a little less.
(2) Please double check the format of all citations.
(3)The paper is quite interesting even if it is advisable to better describe the aims and simulation results of the tests carried out.
Reviewer 5 Report
The authors propose a new planning algorithm based on the RRT named Reduce-Judge RRT (RJ-RRT). The authors need to improve substantial simulation and write up improvements to ensure the manuscript is up to par for publication. First, the authors only compared their algorithm with the basic RRT and RRV. This comparison is not sufficient to prove the scientific contribution of the proposed algorithm. An improvement of an existing algorithm must be compared with similar -RRT-based algorithms that have been published in order to justify the proposed model as an improvement and advancement in knowledge. Surely, when comparing a baseline algorithm, the proposed algorithm generally will have better performance, yet, is this proposed model has an added advantage when compared to other modified/improved RRT algorithms developed for narrow passage applications. Please find a few published works listed here (the majority not cited in this manuscript) that can be used to support your findings. Please compare these models' algorithms on your case study and submit the performance results to clearly support your novelty significance.
- https://www.tandfonline.com/doi/abs/10.1163/016918610X496928
- https://ieeexplore.ieee.org/stamp/stamp.jsp?arnumber=8468190
- https://www.mdpi.com/2076-3417/11/24/11777/htm
- https://ieeexplore.ieee.org/abstract/document/6907540
Then, the authors must show the differences between the length of the near-optimal path and the length of the optimal path (set to 5%, 10%, 15%, 20%, and 25%) or at least a few percentages.
Also, please discuss the results with comparison and revise the findings accordingly.
Round 2
Reviewer 3 Report
The manuscript has been corrected considering the reviewer's remarks and in the current form might be suitable for publication in Applied Sciences journal.